# The Relationship Between Soil and Gut Microbiota Influences the Adaptive Strategies of Goitered Gazelles in the Qaidam Basin

**DOI:** 10.3390/ani14243621

**Published:** 2024-12-15

**Authors:** Yiran Wang, Bin Li, Bo Xu, Wen Qin

**Affiliations:** 1State Key Laboratory of Plateau Ecology and Agriculture, Qinghai University, Xining 810016, China; w18609783019@126.com; 2School of Ecological and Environmental Engineering, Qinghai University, Xining 810016, China; 3Key Laboratory of Adaptation and Evolution of Plateau Biota, Northwest Institute of Plateau Biology, Chinese Academy of Sciences, Xining 810008, China; libin@nwipb.cas.cn (B.L.); xubo@nwipb.cas.cn (B.X.)

**Keywords:** goitered gazelles, soil-derived microorganism, ecological assembly process, source tracking analysis

## Abstract

This study investigates the influence of soil microbiota from different regions on the gut microbiota of goitered gazelles and its role in their adaptation to extreme environments. Using non-invasive sampling methods, we analyzed the composition and diversity of gut and soil microbiota in goitered gazelles across six regions in the Qaidam Basin, Qinghai Province, China. Employing source tracking and ecological assembly process analyses, this is the first study to explore microbiota variations in goitered gazelles across multiple regions. Our findings reveal that soil-derived microbiota significantly influence the composition and diversity of gut microbiota in goitered gazelles, shaping their adaptive strategies. The interaction between gut and soil microbiota plays a crucial role in enhancing the gazelles’ ability to adapt to diverse and extreme environments. Notably, the utilization pattern of soil microbiota by gut microbiota does not correspond to regional trends in gut microbiota α-diversity, potentially due to variations in environmental pressures and the gut filtering capacities of goitered gazelles. These animals actively acquire bacteria from soil microbiota, with the proportion influenced by the state of their gut microbiota. This study underscores the crucial role of gut microbiota in wildlife evolution and highlights the intricate interactions between gut and soil microbiota. By providing new insights into the co-evolution of herbivores and their environments, our findings contribute to a deeper understanding of the adaptive mechanisms in wild herbivores.

## 1. Introduction

Gut microbiota play a critical role in the adaptation of ungulates, particularly those inhabiting extreme environments [1,2,3]. The goitered gazelle, *Gazella subgutturosa* (Güldenstaedt, 1780), primarily found in desert and semi-arid regions, was classified as an endangered species (EN) by the IUCN (International Union for Conservation of Nature) in 2017 due to climate change and habitat loss [4,5]. On the Qinghai–Tibet Plateau, the goitered gazelle is confined to the Qaidam Basin, a region characterized by cold winters and dry summers, and elevations ranging from 2600 to 5500 m. The basin also features saline–alkali soils [6], and vegetation dominated by drought-resistant and salt-tolerant species such as those from the *Asteraceae*, *Amaranthaceae*, and *Gramineae families*, forming distinct desert ecosystems [7]. The Qaidam Basin’s complex geographical environment includes diverse vegetation types: the eastern edge is marked by grassland deserts and mountain grasslands; the western region consists of wind-eroded residual hills with sparse shrub and semi-shrub deserts; the surrounding areas feature mountain deserts and mountain grasslands; and the central plains are dominated by shrub and small semi-shrub deserts, saline thickets, salt marshes, and salt crusts [8]. As a representative herbivore in the Qaidam Basin, goitered gazelles play a crucial role in maintaining ecological balance. Compared to other wild herbivores, they demonstrate exceptional adaptability to alpine, arid, and saline conditions. Previous studies on the adaptability of goitered gazelles have largely focused on populations in the Xinjiang Uygur Autonomous Region, addressing aspects such as diet [9], population structure [10], habitat selection [11], activity rhythms [12,13], and behavioral patterns [14]. In the Qaidam Basin, goitered gazelles rely on their gut microbiota to maintain energy balance, even under nutrient-deficient conditions [15]. Their gut microbiota exhibits significant seasonal variations, with adaptive adjustments in the relative abundance of microbial groups such as *Firmicutes*, *Christensenellaceae*, and *Bacteroides*. Additionally, seasonal changes in soil microbiota directly influence the diversity of the gazelles’ gut microbiota [16]. By selectively ingesting soil microbiota, goitered gazelles can adapt to seasonal environmental fluctuations in the Qaidam Basin. These findings underscore the critical role of soil microbiota in enabling goitered gazelles to cope with the extreme environmental challenges of this region.

Environmental factors play a crucial role in shaping gut microbial diversity, with diet, soil characteristics, and soil microbiota being primary contributors [17,18]. Among these, diet has been extensively studied and is recognized as a key factor influencing the composition and diversity of gut microbiota [19]. Different dietary habits can lead to substantial changes in gut microbiota [20,21]. Soil characteristics, including nutrient status, pH, and texture, impact on soil microbiota [22]. Notably, a reduction in soil organic carbon leads to a decline in soil microbial diversity and diminishes soil fertility. Most bacteria and fungi have been shown positively correlated with soil organic carbon content [23]. While soil microbiota is often overlooked, it plays an important role in animal health, nutrient absorption, and environmental adaptation [24,25,26]. Research on soil characteristics related to ungulates remains scarce, highlighting a critical gap in understanding how soil and microbiota interactions influence these species. This gap is particularly significant given that many herbivorous mammals do not form local populations uniformly; instead, they tend to establish themselves primarily in specific types of soil [27,28]. Soil microbiota can significantly alter the composition of gut microbiota in mice, and non-sterilized soil further promotes increased bacterial diversity [24]. Furthermore, early exposure to soil microbiota facilitates the establishment of beneficial gut microbiota in mammals, increasing the ratio of *Bacteroidetes* to *Firmicutes* and the abundance of *Ruminococcaceae*, contributing to overall health [25,26]. For wild herbivores, such as goitered gazelles, the interaction with soil microbiota occurs naturally during feeding. However, previous source tracking analyses indicated that the impact of soil microbiota on the gut bacterial communities of other mammals, such as zokors (approximately 1% ASV from soil) [29], Tibetan antelopes (0.75% ± 0.60%), and Przewalski’s gazelles (3.15% ± 4.94%) [30] is relatively minimal. In contrast, our study demonstrated a significantly higher proportion of soil-associated microbiota (PSM) in the gut microbiota of goitered gazelles, exceeding 10% [16]. This suggests that soil plays a more pronounced role in shaping the gut microbiota of goitered gazelles compared to other herbivores, including rodents and Tibetan antelopes. While diet is widely acknowledged as the primary factor influencing gut microbiota, our observations indicate that goitered gazelles (*Gazella subgutturosa*) in the Qaidam Basin occasionally lick soil during feeding. The underlying reasons for this behavior remain unclear. We propose two hypotheses: (1) goitered gazelles, like other livestock, may deliberately lick soil to supplement essential nutrients; and (2) soil licking may occur unintentionally during feeding, resulting in the ingestion of soil microorganisms. However, under either hypothesis, the effects of soil microorganisms on the gut microbiota of goitered gazelles remain unknown. Based on these premises, this study focuses exclusively on the relationship between soil microbiota and the gut microbiota of goitered gazelles. This finding underscores the potential importance of soil-derived microorganisms in shaping the gut microbial communities of herbivores.

Using non-invasive sampling, we analyzed the composition and diversity of gut microbiota in goitered gazelles across six key locations in the Qaidam Basin. The aim of this study was to investigate how soil microbiota communities in different regions affect the composition and diversity of gut microbiota communities in goitered gazelles, especially how goitered gazelles regulate their gut microbiota by acquiring microbiota from the soil to adapt to extreme environments under different environmental stresses. This study highlights the pivotal role of soil-derived microorganisms in shaping gut microbial communities in wild herbivores, revealing their importance in adaptive strategies. Furthermore, our findings provide novel insights into the interactions between gut and soil microbiota in herbivores, their ecological impacts, and their roles in evolutionary processes. This research advances our understanding of the co-evolutionary mechanisms between wildlife and their environments, emphasizing the integral role of microbiota communities in ecological adaptation.

## 2. Materials and Methods

### 2.1. Sample Collection

A total of 6 sampling points were chosen in the Qaidam Basin, Qinghai province, China, and the sampling was conducted from 9 March to 18 March 2023. The choice of winter sampling was due to the scarcity of food in winter, which can better reveal whether goitered gazelles depend on soil microbiota. Figure 1 provides a visual representation of goitered gazelles and their habitat, which is essential for understanding the context of the sampling points discussed. The sampling points were as follows: Keke (KK), Haersi (HES), Geermu (GEM), Gangci (GC), Huaitoutala (HTTL), and Keluke (KLK) (Figure 2). At each site, we collected 5–8 fresh fecal samples of goitered gazelles and the corresponding soil samples, resulting in a total of 74 samples (38 feces samples and 36 soil samples). During fieldwork, samples were placed in liquid nitrogen after labeling, and then transferred to a −80 °C refrigerator before the experiment.

### 2.2. DNA Extraction, Amplification, and Sequencing

The library preparation and sequencing were performed at Biomarker Technologies Co., Ltd. (Beijing, China). DNA extraction from soil and fecal samples was conducted using the TGuide S96 magnetic bead-based soil/fecal genomic DNA extraction kit (TIANGEN, Beijing, China). To ensure consistency, both soil and fecal samples were processed using the same reagent kit. For soil samples, 0.25–0.5 g of soil was combined with 500 μL of SA buffer, 100 μL of SC buffer, and 0.25 g of turmeric in a 2 mL centrifugal tube and then centrifugated. The mixture was homogenized using a TGrinder H24 tissue homogenizer (OSE-TH-01, TIANGEN, Beijing, China). For samples with low yield, vortex mixing or homogenization was performed, followed by heating at 70 °C for 15 min to enhance lysis efficiency. After adding the fecal samples, an appropriate buffer, and grinding beads to the 2 mL centrifuge tube, a similar processing step was performed. For fecal samples which may contain residual RNA, 10 µL of RNase A (TIANGEN, Beijing, China) was added. For Gram-positive bacteria with cell walls, which are difficult to lyse, the lysis temperature was increased to 95 °C. Subsequent steps included the addition of SH buffer, centrifugation, GFA buffer, magnetic bead suspension G, incubation on a magnetic stand, and treatment with deproteinized liquid RD. DNA solution was transferred to a new centrifuge tube and eluted using TB buffer.

DNA concentration was measured using a microplate reader. The integrity of PCR products was verified by electrophoresis on a 1.8% agarose gel. The complete 16S rRNA gene was amplified with primers 27F_(16S-F) (5′-AGR GTT TGA TYN TGG CTC AG-3′) and 1492R_(16S-R) (5′-TAS GGT ACG TTT ACG ACT T-3′) [31]. The 30 μL PCR reaction contained 1.5 µL of genomic DNA, 10.5 µL of NFW (Nuclease-Free Water), 15 µL of KOD ONE MM (KOD OneTM PCR Master Mix, Bailingke, Beijing, China), and 3 µL of barcode primers. The PCR program was as follows: 95 °C for 2 min, 98 °C for 10 s, annealing at 55 °C for 30 s, and DNA chain extension at 72 °C for 90 s, repeated for 25 cycles. A final extension step at 72 °C for 2 min ensured the complete synthesis of all DNA strands [32,33]. Amplification products were concentrated and checked. The mixed products underwent damage repair, end repair, and adapter ligation with the SMRTbell Template Prep Kit (Pacific Biosciences, Menlo Park, CA, USA), and the reaction was performed using PCR instrument. The final library was recovered with AMpure PB magnetic beads. Before sequencing, the library was bound to primers and polymerase using the PacBio Binding kit (Pacific Biosciences, Menlo Park, CA, USA). After a purification with AMpure PB beads, all samples were sequenced on a Sequel II (Pacific Biosciences, Menlo Park, CA, USA). A total of 74 samples were analyzed.

### 2.3. Bioinformatics Methods

We used SMRT Link v8.0 software [34] to correct the raw PacBio sequencing data and generated Circular Consensus Sequencing (CCS) reads with an accuracy exceeding 99%. Barcode identification was performed using lima v1.7.0 software (https://lima-vm.io, accessed on 7 September 2023.) to assign CCS reads to different samples, yielding Raw-CCS read data. Subsequently, Cutadapt v1.9.1 [35] was employed to remove primer sequences and perform length filtering (1200–1650 bp), resulting in Clean-CCS reads. UCHIME v4.2 [36] was employed to identify and remove chimeric sequences, yielding Effective-CCS reads. For subsequent analyses, Usearch v10.0 [37] was used to cluster reads at a 97.0% similarity threshold, with operational taxonomic units (OTUs) [38] filtered using a default threshold of 0.005% of the total sequences count. This process resulted in OTUs, and each OTU was represented by a unique sequence. Diversity analysis, differential analysis, and source tracking were performed using the SILVA database (Release138, http://www.arb-silva.de, accessed on 7 September 2023) [39] and the RESCRIPt tool [40]. We performed species annotation with a combined approach in QIIME2 [41], using classify-consensus-blast (alignment-based) and classify-sklearn (naive Bayes classifier), with a standard threshold of 70%. The above steps were performed by Biomarker Technologies Co., Ltd. (Beijing, China). OTUs classified as mitochondria and chloroplasts were removed. Subsequently, the OTU (the rarefied OTU table shown in the Appendix A) table for all samples was rarified to the minimum sequencing depth (4351) for downstream analyses, and some OTUs with values of 0 in all samples needed to be culled as the total OTU abundance was reduced, except for the FEAST analysis [42]. FEAST analysis required the original OTU table. The Venn diagrams were generated using the “VennDiagram-v1.6.9” package [43]. The pairwise comparisons of microbiota between gut and soil samples at the phylum, family, and genus levels were calculated based on the Wilcoxon rank sum test. Alpha diversity indices (Shannon and Chao1) were calculated using the “picante-v1.8.2” package [44], and compared by the Wilcoxon rank sum test with the “stats-v4.4.1” package between any two groups [45]. Beta diversity analyses were based on Bray–Curtis distances, which was calculated by the “vegan-v2.4.3” package [46]. Non-metric multidimensional scaling (NMDS), analysis of similarities (ANOSIM) [47], and permutational multivariate analysis of variance (PERMANOVA) [48] were performed using the “vegan” package with 999 permutations [46] and visualized using the package “ggplot2-v3.5.1” [49].

### 2.4. Source Tracing Analysis

The “FEAST” (Fast Expectation-Maximization Microbial Source Tracking) package was employed in Rstudio v4.4.1 to elucidate the origin of the goitered gazelle gut microbiota [42]. In this analysis, the gut microbiota of goitered gazelles was designated as the “sink”, while the soil microbiota was considered the “source”. Gut microbiota and soil microbiota were compared according to different regions, and unannotated taxa were classified as “unknown”. In the “FEAST” analysis, all parameters were set to EMiterations = 100,000.

### 2.5. Ecological Assembly Process of Soil and Gut Microbiota

We used the Modified Stochastic Ratio (MST) to estimate the contributions of stochastic and deterministic assembly processes in shaping the gut and soil microbiota. MST values were calculated by the “NST” (Normalized Stochastic Ratio) package in R v4.4.1 and RStudio based on 30,000 simulations [50,51]. MST values above and below 0.5 represent stochastic and deterministic assembly processes, respectively [50]. We adhered to the methodology proposed by the authors of the NST package (https://github.com/DaliangNing/NST, accessed on 2 August 2024).

## 3. Results

### 3.1. Raw Data

Sequencing and barcode identification of the 74 samples yielded a total of 942,974 CCS reads. Each sample generated at least 8490 CCS reads, with an average of 12,743 CCS reads per sample. The average length of the full 16S rRNA gene was 1448.68 bp. Clustering analysis identified a total of 6539 OTUs across all samples. The number of OTUs per sample varied from 338 to 1309. A total of 682 OTUs were shared among the gut microbiota of goitered gazelles from six regions. The GC group showed the highest number of unique OTUs (172), followed by the HES (143) and KLK (133) groups. The GEM group showed the lowest number of unique OTUs (60) (Figure 3A). In the soil microbiota, 44 OTUs were shared among six groups. The HTTL group demonstrated the highest number of unique OTUs (357), followed by KLK (237), and the KK group had the lowest number of unique OTUs (125) (Figure 3B).

### 3.2. Composition and Diversity of Soil and Gut Microbiota

In the gut microbiota of goitered gazelles, the top five phyla in relative abundance were *Firmicutes*, *Bacteroidota*, *Proteobacteria*, Cyanobacteria, and *Verrucomicrobiota*. All phyla except the *Verrucomicrobiota* showed significant differences among the six groups (*p* < 0.05, Figure 4A). At the family level, the top five families were *Rikenellaceae*, *Lachnospiraceae*, *Oscillospiraceae*, *Christensenellaceae*, and *[Eubacterium]_coprostanoligenes_group*, and only the relative abundance of *Lachnospiraceae* and *[Eubacterium]_coprostanoligenes_group* showed no significant differences among the six groups (*p* > 0.05, Figure 4B). At the genus level, the top five bacteria were *Rikenellaceae_RC9_gut_group*, *UCG-05*, *Christensenellaceae_R-7_group*, *uncultured_rumen_bacterium*, and *unclassified_Lachnospiraceae*. Only the relative abundance of *Rikenellaceae_RC9_gut_group* and *unclassified_Lachnospiraceae* showed no significant differences among the six groups (*p* > 0.05, Figure 5A).

In the soil microbiota, the top five phyla were *Bacteroidota*, *Proteobacteria*, *Acidobacteriota*, *Actinobacteriota*, and *Firmicutes*, all of which showed significant differences (*p* < 0.05; Figure 6A). At the family level, the top five families were *Flavobacteriaceae*, *Balneolaceae*, *Halomonadaceae*, *Hymenobacteraceae*, and *Rhodobacteraceae*. Only the relative abundances of *Hymenobacteraceae* and *Rhodobacteraceae* showed no significant differences (*p* > 0.05; Figure 6B). At the genus level, the top five genera were *Salinimicrobium*, *Unclassified_f_Balneolaceae*, *Antarcticibacterium*, *Salegentibacter*, and *Halomonas*, all of which showed significant differences (*p* < 0.05; Figure 6C).

For gut microbiota, according to the Shannon and Chao1 indices, the α-diversity of gut microbiota in the GC groups had the lowest values at the OTU level (*p* < 0.05). The KLK, HES, and HTTL groups exhibited the highest α-diversity, with no significant differences among these three groups (*p* > 0.05, Figure 7A,B). For soil microbiota, at the OTU level, according to the Shannon and Chao1 indices, the GC group showed the lowest α-diversity (*p* < 0.05), while the HTTL group exhibited the highest diversity, and a significant difference was observed between the HTTL and GC groups (*p* <0.05; Figure 8A,B).

In terms of β-diversity, PERMANOVA analysis revealed significant differences in gut microbiota diversity among the six groups (R^2^ = 0.323, *p* = 0.001). ANOSIM analysis at the OTU level (R^2^ = 0.666, *p* = 0.001) also indicated that inter-group differences were significantly greater than intra-group differences. The NMDS analysis indicated that the gut microbiota of the goitered gazelle from the HES and KK groups exhibited higher similarities (*p* > 0.05), while the gut microbiota from the GC group was distinctly separated from those of other groups (*p* < 0.05) (Figure 9A). For soil microbiota, PERMANOVA results showed significant differences among the six groups (R^2^ = 0.766, *p* = 0.001), and ANOSIM indicated that between-group differences were higher than within-group differences (R = 0.95, *p* = 0.001). The NMDS analysis revealed a clear separation between the GC group and the other five groups (*p* < 0.05) (Figure 9B).

### 3.3. Gut Microbial Source Tracking from Soil Microbiota

Source tracking analysis revealed varying contributions of soil microbiota to gut microbiota across different groups. The source tracking results showed the lowest contribution in HTTL and the highest in GC. The GC group showed the highest average contribution rate (8.94%, range: 6.26–12.20%), while the HTTL group exhibited the lowest (1.80%, range: 0.54–3.11%). For the KK, KLK, HES, and GEM groups, the average contribution rates are 5.43%, 2.88%, 2.47%, and 2.08%, respectively (Table 1). The pairwise comparisons indicated significant differences between the GC group and the other five groups (*p* < 0.05). Additionally, significant differences were observed between the KK and HES, HTTL, and GEM groups, respectively (*p* < 0.05), However, no significant differences were observed among the three groups, HTTL, HES, and KLK, in pairwise comparisons (*p* > 0.05).

### 3.4. Dominant Ecological Processes in Gut and Soil Microbiota

MST analysis indicated that the dominant ecological process of gut microbiota was stochastic (MST > 0.5), the dominant ecological process of soil microbiota was deterministic (MST < 0.5), with the exception of the KLK group, where stochastic processes were dominant (MST > 0.5) (Figure 10).

## 4. Discussion

### 4.1. Differences in Soil and Gut Microbial Diversity Across Six Regions

The analysis of α-diversity revealed that both gut and soil microbiota in the GC region exhibited the lowest diversity. We hypothesize that this may be attributed to the fact that the Qaidam Basin exhibits significant regional differences in vegetation diversity, and the GC region has fewer plant species, mainly including *Ceratoides latens* and *Tamarix arceuthoides*. The lowest α-diversity in both gut and soil microbiota of GC may be related to the severe desertification, poor soil quality, and limited vegetation diversity [7,52,53]. Additionally, during sampling, we observed that the soil in the GC had higher salinity and alkalinity levels, which significantly affect soil microbial communities [54] and may further contribute to the low soil microbial diversity. A lower gut microbiota diversity of GC may help minimize the host’s energy expenditure in maintaining microbial stability, thereby conserving energy in such resource-poor environments.

In contrast, both α-diversity in gut and soil microbiota were highest in the HTTL, which may be closely linked to the greater vegetation diversity of this area. In the HTTL, more diverse plant communities are supported, including various shrub species (primarily consisting of semi-arid and shrub desert types) [9,53]. Higher richness in vegetation may increase the α-diversity of local soil microbiota [55]. Consequently, the α-diversity of soil microbiota in HTTL is the highest. Vegetation diversity directly influences soil microbial communities, which in turn may indirectly influence the gut microbiota of herbivores such as goitered gazelles. Richer vegetation likely supports more diverse soil microbial communities. More diverse plant communities may provide a wider range of nutritional resources for goitered gazelles, to maintain the homeostatic balance within its gut.

### 4.2. Variations in Source Tracking Results Linked to Host Needs

According to the source tracking analysis, the proportion of microbiota acquired from soil was highest in the GC, an unexpected finding. We hypothesize that under the conditions of low gut microbial α-diversity in the GC, goitered gazelles may rely on external sources of microorganisms to increase diversity, thereby maintaining certain gut functions and preserving intestinal homeostasis. In the resource-poor environment of the GC, where vegetation and food availability are limited, goitered gazelles may be unable to obtain sufficient nutrients from their diet to sustain gut microbial α-diversity. Consequently, they may rely on soil as an alternative source of microbial populations. In contrast, the proportion of soil-derived microorganisms was lowest in the HTTL. This may be attributed to the region’s rich vegetation and abundant food resources. Although higher vegetation richness is known to enhance local soil microbial α-diversity [55], goitered gazelles in the HTTL region likely obtain sufficient nutrients from their diet to sustain a higher level of gut microbial α-diversity, reducing their need to acquire microorganisms from soil.

This relationship suggests a cascading effect where vegetation diversity directly influences soil microbial communities, which in turn may indirectly influence the gut microbiota of herbivores such as goitered gazelles. Richer vegetation likely supports more diverse soil microbial communities. More diverse plant communities may provide a wider range of nutritional resources for goitered gazelles, potentially reducing their reliance on soil ingestion for mineral or microbial supplementation. The reduced soil ingestion in areas with richer vegetation could lead to a lower transfer of soil microbiota to the gut microbiota. Therefore, we hypothesized that the richer plant resources in HTTL enable goitered gazelles to obtain sufficient nutrients from vegetation, reducing their need to ingest soil. Conversely, gazelles in the GC region, faced with scarcer food resources, may resort to more frequent soil ingestion while foraging, increasing their exposure to soil microbiota. The proportion of microorganisms acquired from soil by goitered gazelles ranged from 1.8% to 8.9%, which is significantly higher compared to other species on the Qinghai–Tibet Plateau [29,30]. Moreover, the source tracking results for different goitered gazelle populations varied significantly across regions, and these variations did not align with the α-diversity patterns observed in gut and soil microbiota. This suggests that the proportion of soil microbiota acquired by goitered gazelles is determined by the state of their gut microbiota. We propose that the consumption of soil microbiota by goitered gazelles is intentional rather than incidental, reflecting a purposeful behavior. The significant differences in soil microbiota intake among goitered gazelle populations across regions may be attributed to habitat variability, which likely influences their specific needs for soil microbial groups and quantities.

This study is consistent with previous source tracking analyses of goitered gazelles [16], which have shown that the average proportion of soil-derived microbiota in their gut is higher than in other species, exceeding 4%. Based on this, we infer that goitered gazelles maintain a relatively stable and substantial reliance on soil microbiota. Compared to Przewalski’s gazelles and Tibetan gazelles [30], goitered gazelles exhibit a significantly higher proportion of gut microbiota derived from soil, highlighting clear interspecies differences. This finding suggests that each species adopts distinct adaptive strategies for utilizing soil microbiota, selectively enriching specific microbes to meet their ecological needs. Additionally, the acquisition of gut microbiota from soil appears to be influenced by environmental factors, further contributing to these interspecies differences. A higher reliance on soil-derived microbiota may also be indicative of lower adaptability. Compared to Tibetan gazelles and Przewalski’s gazelles, the greater proportion of soil microbiota observed in goitered gazelles could reflect reduced adaptive capacity. Alternatively, it may be a response to the harsh environmental conditions and limited vegetation diversity of the Qaidam Basin, where goitered gazelles primarily reside. Further research is needed to validate these hypotheses and explore the underlying mechanisms.

### 4.3. Differences in Ecological Processes of Soil and Gut Microbiota

The assembly processes of microorganisms reflect the mechanisms underlying microbial community composition [56,57]. In the gut microbiota, ecological processes are predominantly stochastic, with community changes driven primarily by phenomena such as microbial birth, death, and the colonization of new microbial taxa [58,59]. The gut microbiota is a highly dynamic system that can rapidly adapt to changes in the host’s physiological state [60]. These neutral mechanisms can lead to shifts in the composition and abundance of gut microbial communities over time, even in the absence of selective pressures from the host environment. Meanwhile, under healthy conditions, the gut microbiota maintains a stable homeostasis [61,62]. In contrast, the ecological processes shaping soil microbiota are largely deterministic, with assembly processes governed by predation, competition, and similar interactions [63,64]. The harsh environmental conditions of the Qaidam Basin allow only specific microorganisms to survive, a result of natural selection. In such an environment, the colonization of selected microorganisms is more advantageous than that of others, favoring their dominance within the community. Our findings largely corroborate the established theory that stochastic processes primarily drive the assembly of animal-associated microbial communities, while deterministic processes play a more significant role in plant-associated microbial communities [58,63].

### 4.4. Diverse Adaptation Strategies of Goitered Gazelles via Soil Microbial Utilization Across Regions

*Firmicutes*, *Bacteroidetes*, *Verrucomicrobiota* and the genera *Akkermansia, Christensenellaceae_R-7_group,* and *Rikenellace-ae_RC9_gut_group* (information on bacteria in soil at the level of the relevant genus has been collated and placed in the Appendix A) are both found in gut and soil microbiota. These collectively suggest that goitered gazelles may obtain these bacteria from soil microbiota.

The main functions of *Firmicutes* and *Bacteroidetes* are to assist the host in digesting and absorbing nutrients, with *Firmicutes* primarily involved in carbohydrate degradation and *Bacteroidetes* in protein degradation [65]. Those microbial functions are essential for the survival of goitered gazelles in the extreme environment of the Qaidam Basin. Among the HES, KLK, GEM, KK, and HTTL regions, the relative abundance of the *Firmicutes* and *Bacteroidetes* showed no significant differences (*p* > 0.05). This similarity in gut microbiota may be correlated with geographic proximity [66,67]. We hypothesize that the geographic closeness of these regions may result in similar soil environments and food resources, leading to comparable soil and gut microbial diversity [68]. This observation underscores the potential influence of landscape-scale environmental factors on the gut microbiome composition of goitered gazelles. The higher proportion of F/B (*Firmicutes*/*Bacteroidetes*) in the gut microbiota is associated with the increased energy extraction efficiency from the diet [69,70]. Interestingly, we observed that the F/B (*Firmicutes*/*Bacteroidetes*) ratio in the GC was significantly higher than that of other groups (*p* < 0.05). We hypothesize that this elevated F/B ratio may represent an adaptive strategy, enabling goitered gazelles in the GC region to more effectively absorb energy from their diet, thus maintaining energy homeostasis under harsh environmental conditions. At the genus level, the relative abundances of *Christensenellaceae_R-7_group* and *Rikenellaceae_RC9_gut_group* were significantly higher in the GC region compared to the HTTL region. *Christensenellaceae_R-7_group* is known to enhance the degradation of fibrous materials and improve plant nutrient utilization. In the GC region, the plant community is dominated by Ceratoides latens, a species with a crude protein content exceeding 10%. This indicates that *Christensenellaceae_R-7_group* plays a crucial role in facilitating food digestion and nutrient acquisition in goitered gazelles from the GC region. Meanwhile, high crude protein levels are associated with a decrease in the relative abundance of *Rikenellaceae_RC9_gut_group*, which is involved in the degradation of hemicellulose and other food components, as well as various metabolic activities. In the gut, *Rikenellaceae_RC9_gut_group* facilitates the degradation of cellulose, while in the soil, it likely contributes to the transformation of soil organic matter and the release of nutrients. This dual presence of the above two genera suggests that these microbial taxa play important ecological roles across different ecosystems, such as promoting organic matter decomposition and nutrient cycling.

We found no significant differences in the *Verrucomicrobiota* and its dominant genus *Akkermansia* among the gut microbiota of goitered gazelle from the six groups (*p* > 0.05). That means the relative abundances of *Verrucomicrobiota* and *Akkermansia* are stable in the gut microbiota of goitered gazelles from the six regions in the Qaidam Basin. *Akkermansia* is known to be promoted by plant secondary compounds in the diet [71,72]. It has also demonstrated probiotic effects, including the preservation of mucus layer thickness, reduction in gut permeability, modulation of systemic lipopolysaccharide levels, and reduction in adiposity and inflammation [72]. Therefore, we speculate that the consistent presence of *Akkermansia* across different regions suggests its importance in maintaining gut homeostasis, promoting intestinal health, and potentially mitigating metabolic disorders in goitered gazelles inhabiting the diverse environments of the Qaidam Basin.

The gut microoganism obtained from soil microbiota of goitered gazelles plays a crucial role in their adaptation to the varied and often harsh environments of the Qaidam Basin. The stability of certain microbial taxa across regions, coupled with region-specific variations in others, may represent a balance between conserved adaptive traits and local environmental adaptations.

## 5. Conclusions

The gut and soil microbiota diversity of goitered gazelle populations in different regions of the Qaidam Basin exhibit significant variation. Goitered gazelles actively acquire microorganisms from soil to maintain the homeostatic balance of their gut microbiota. However, due to substantial differences in environmental conditions and food resources across regions, the proportion of soil microorganisms acquired by goitered gazelles varies. Populations in harsher environments rely more heavily on soil microbiota, while those in regions with more favorable conditions have a lower dependence. The microorganisms obtained from soil play a crucial role in supporting the survival, metabolism, and nutrient absorption of goitered gazelles.

## Figures and Tables

**Figure 1 animals-14-03621-f001:**
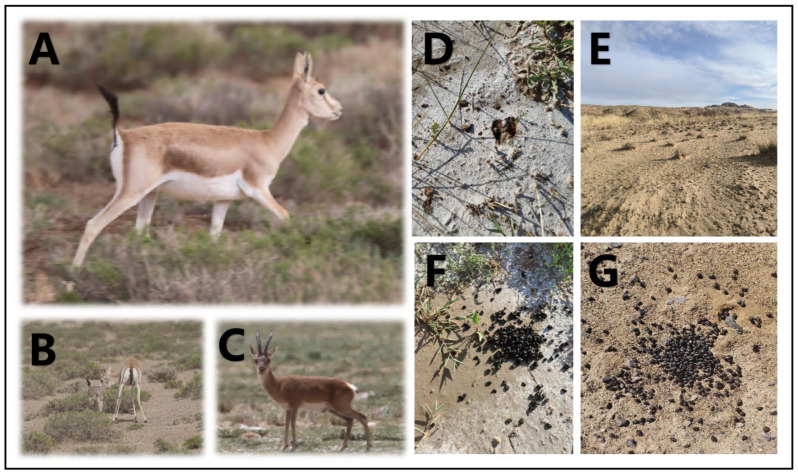
Photographs of (**A**) goitered gazelles in side view, (**B**) goitered gazelles with offspring, (**C**) goitered gazelles in frontal view, (**D**) hoofprints of goitered gazelles, (**E**) desert landscapes in Wulan Town, and (**F**,**G**) fresh feces of goitered gazelles.

**Figure 2 animals-14-03621-f002:**
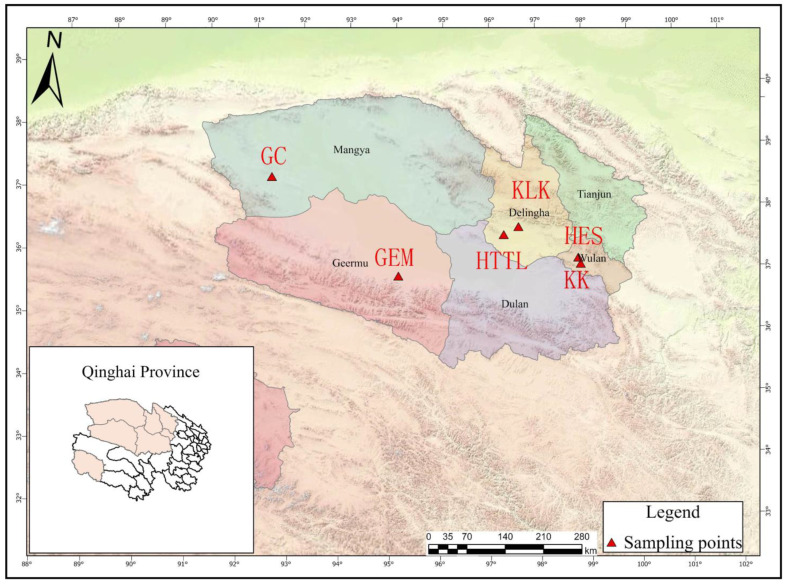
Sampling points in the Qaidam Basin, Qinghai province, China (KK: Keke, 5 fecal samples. HES: Haersi, 8 fecal samples. KLK: Keluke, 5 fecal samples. HTTL: Huaitoutala, 6 fecal samples. GEM: Geermu, 5 fecal samples. GC: Gangci, 9 fecal samples).

**Figure 3 animals-14-03621-f003:**
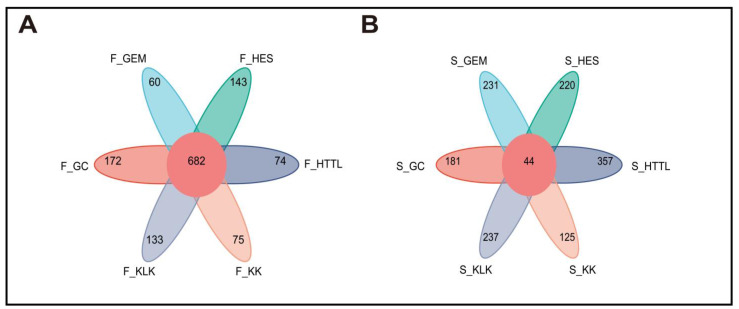
Venn diagram among 6 regions of (**A**) gut microbiota and (**B**) soil microbiota at the OTU level (F represents fecal; S represents soil).

**Figure 4 animals-14-03621-f004:**
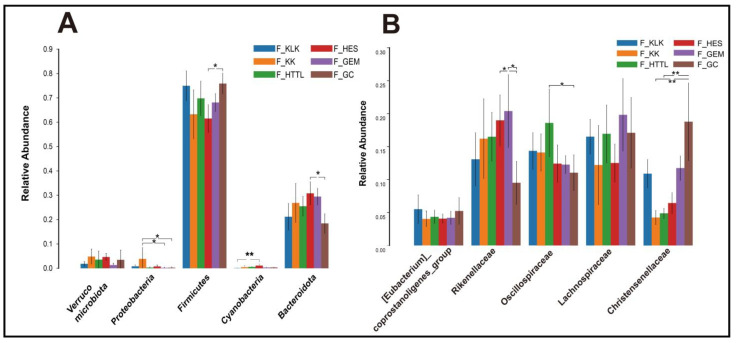
The top 5 phyla (**A**) and families (**B**) in relative abundance of gut microbiota among 6 regions based on the Wilcoxon rank sum test (F represents fecal; * represents *p* < 0.05; ** represents 0.001 < *p* < 0.01).

**Figure 5 animals-14-03621-f005:**
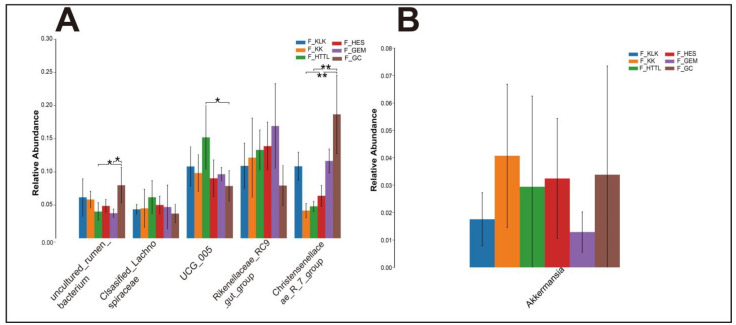
The top 5 genera (**A**) in relative abundance and (**B**) *Akkermansia* of gut microbiota among 6 regions based on the Wilcoxon rank sum test (F represents fecal; * represents *p* < 0.05; ** represents 0.001 < *p* < 0.01).

**Figure 6 animals-14-03621-f006:**
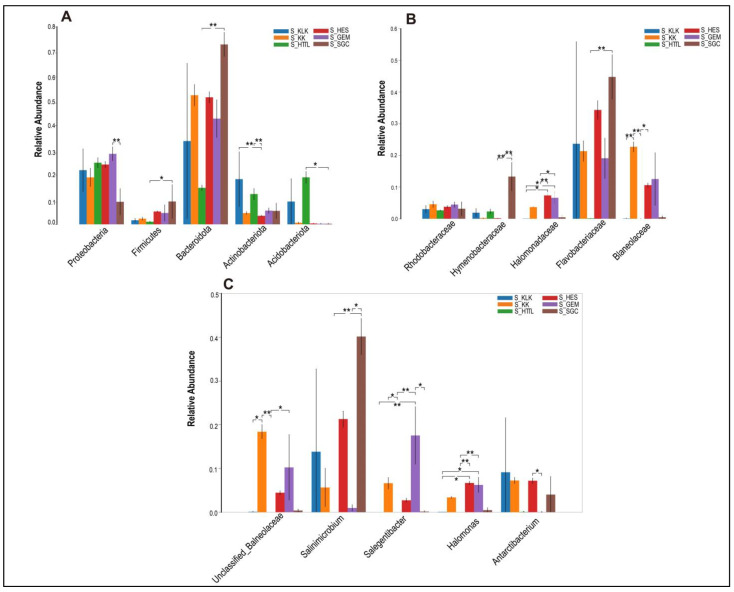
The top 5 phyla (**A**), families (**B**) and genera (**C**) in relative abundance of soil microbiota among 6 regions based on the Wilcoxon rank sum test (S represents soil; * represents *p* < 0.05; ** represents 0.001 < *p* < 0.01).

**Figure 7 animals-14-03621-f007:**
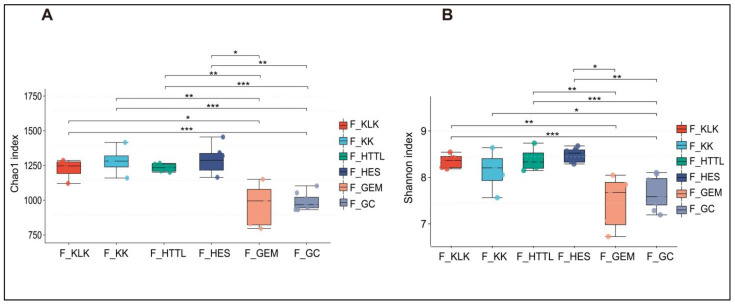
The α-diversity of the Chao1 index (**A**) and Shannon index (**B**) in gut microbiota at the OTU level among 6 regions based on the Wilcoxon rank sum test (F represents fecal; * represents *p* < 0.05; ** represents 0.001 < *p* < 0.01; *** represents *p* < 0.001).

**Figure 8 animals-14-03621-f008:**
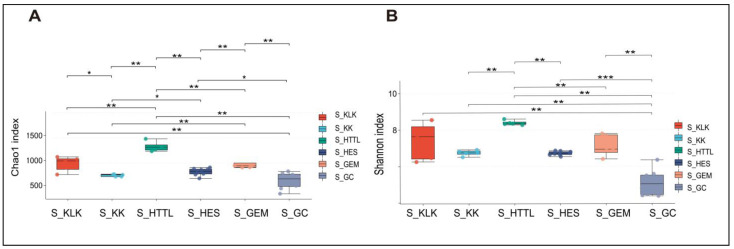
The α-diversity of Chao1 index (**A**) and Shannon index (**B**) in soil microbiota at the OTU level among 6 regions based on the Wilcoxon rank sum test (S represents soil; * represents *p* < 0.05; ** represents 0.001 < *p* < 0.01; *** represents *p* < 0.001).

**Figure 9 animals-14-03621-f009:**
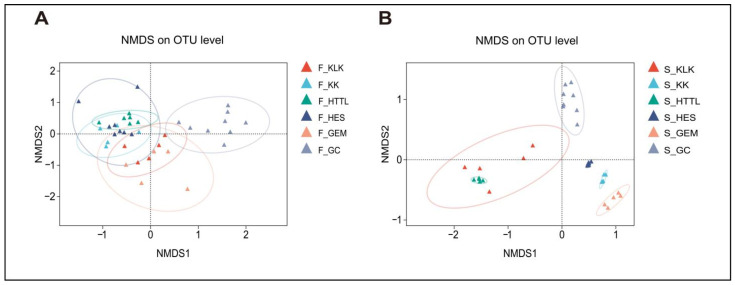
NMDS analysis among 6 regions of (**A**) gut microbiota and (**B**) soil microbiota at the OTU level based on Bray–Curtis distance matrices (F represents fecal; S represents soil).

**Figure 10 animals-14-03621-f010:**
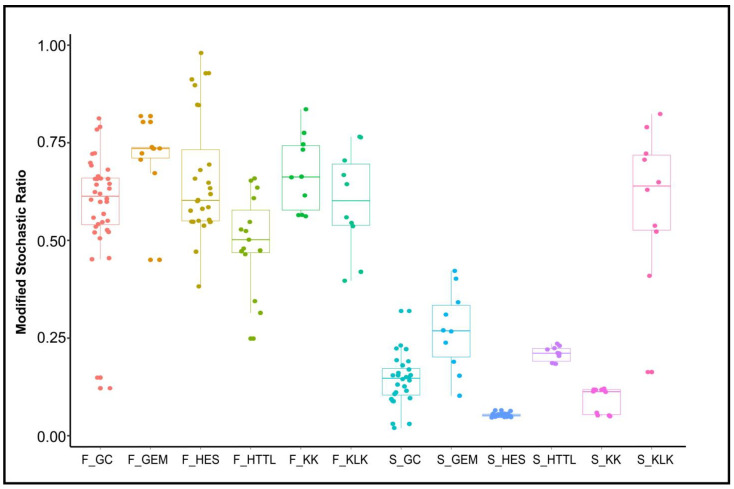
Analysis of Modified Stochastic Ratio (MST) of gut and soil microbiota (F represents fecal; S represents soil).

**Table 1 animals-14-03621-t001:** Results of source tracking analysis in different regions.

Group	Soil Contribution Range/%	Soil Contribution Average/%
GC	6.26–12.20	8.94 (±2.97)
GEM	1.29–2.92	2.08 (±0.82)
HES	1.17–3.49	2.47 (±1.16)
HTTL	0.54–3.11	1.80 (±1.29)
KK	2.60–8.96	5.43 (±3.18)
KLK	1.33–5.42	2.88 (±2.05)

## Data Availability

Raw reads of 16S rRNA sequencing data have been submitted to Sequence Read Archive (SRA) under the BioProject ID PRJNA1105749 in NCBI.

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
