# Peer review of "The Relationship Between Soil and Gut Microbiota Influences the Adaptive Strategies of Goitered Gazelles in the Qaidam Basin"

_animals, 2024, doi:10.3390/ani14243621_

Round 1
Reviewer 1 Report
Comments and Suggestions for Authors
Dear Authors,
The manuscript certainly touches upon an interesting topic. The influence of soil microbiota on the intestinal microbiota of herbivorous mammals is an important topic of research with applied significance. Gazelle goitered gazelles are mammals for which there is little information in this regard. However, the manuscript cannot be published in its current form. The title of the manuscript needs to be adjusted, since the soil microbiota is primary. The text of the manuscript is mixed up in different chapters and should be moved to the appropriate chapters. I was particularly confused by the lack of Discussion. It is in the Results to some extent, but it is not a Discussion. It should be expanded in comparison with the study of other herbivores. The manuscript pays very little attention to vegetation, but it is a transitional component from the soil to the intestine of the goitered gazelle. A hypothesis needs to be formulated. The authors obtained interesting results, presented them, and based on the proposed hypothesis, they should be disclosed in the conclusions of the manuscript. After all the comments have been corrected, the manuscript can be reviewed again.

Reviewer 2 Report
Comments and Suggestions for Authors
The paper of Wang et al., “The relationship between gut and soil microbiota influences the adaptive strategies of goitered gazelles in the Qaidam basin” is devoted studying how soil microbiota from different regions influences the gut microbiota of goitered gazelles and its role in adapting to extreme environments. This is a pretty interesting article overall. However, there are questions and comments that need to be answered.
Introduction
1. Lines 32-33. The first occurrence of GC and HTTL should be polished.
2. Lines 82-26. “This study is significant for understanding how gut microbiota help goitered gazelles adapt to the extreme environmental conditions of the Qaidam Basin, thereby enhancing their survival in these harsh habitats. This discovery not only enriches the theory of the application of microbiota in ecological adaptation, but also provides a new scientific basis for the conservation and management of endangered species”. I do not think so, and this prediction is global. I do not see any connections between your results and this conclusion.
3. The main question: Why did the authors not study the relationship between the gut and plant (mainly phyllosphere) microbiota instead of the relationship between the gut and soil microbiota. Because the flora that serves as food for goitered gazelles are the main sources of microbiota than the soil microbiota.
Materials and Methods
4. It is quite possible that fecal samples from one animal will be collected twice, as stages of goitered gazelles can migrate from one territory to another during February 8-18. How was this issue resolved?
5. Figure 1, 2. Abbreviations of territories should be polished.
6. Line 153. Why were archaeal OTUs removed? Archaea are part of the microbiota and their role in the ruminant gut is significant. Please bring back archaea and reanalyze the data including archaea.
Results
7. Line 91. Raw Data.
8. Lines 192-193. I know that the number of 16S rRNA amplicons varies for each sample and they need to be normalized before clustering the reads. Did you normalize the number of reads?
9. Line 269. 3.3. Gut Microbial Source Tracking from Soil Microbiota. The tool you used for the microbiota source tracing analysis may give relative results. However, it does not say anything about which bacteria or archaea originating from the soil are ultimately found in the goitered gazelle fecal microbiome. I would like to see which specific bacteria are preserved? Without assessing the composition of the microbiota, the results of such an analysis are somehow meaningless.
Discussion
10. After answering to my questions, the Discussion part will be changed, undoubtedly.
11. Line 298. Did you want to write Discussion instead of Results?
12. Line 413. Throughout the text of the article, provide the composition of the microbiota at the phylum level. At such a high level, it is completely impossible to judge where and how and what changes. Therefore, I advise you to describe the composition of the microbiota at a lower taxonomic level.
13. You have previously conducted a similar study and you have studied fecal samples of goitered gazelles from the same province. I am sure that other researchers have conducted similar studies besides you. And the results of these studies should be properly compared with yours and discussed in the discussion.
14. Lines 439-440. Our research contributes to the understanding of how wild herbivores adapt to extreme environments through gut microbiome modulation. I think other results were needed to reach this conclusion. Unfortunately, you were unable to show how wild herbivores adapt to extreme environmental conditions by modulating the gut microbiome. Your research is about something else entirely and you do not have results on the part of the soil microbiota that could participate in modulating the goitered gazelle gut microbiome. Please analyze what is written.
15. As additional material, please provide the OUT table. But please note that if you provide the OUT table, the data provided in it should be carefully presented in the text of the article, i.e. if, for example, there is a dominant OTU in the table, it should be mentioned in the article.
16. There are a lot of illustrations. I don't think they are all necessary and you should leave those that are being discussed.
General conclusion. The data you obtained is new and I do not want this data to disappear, so I decided to send your article for major revision instead of rejecting it. However, the article lacks the approach of a professional microbiologist. And many of the results obtained do not logically fit with the statements and conclusions. It is necessary to clearly formulate the significance of the results obtained. Therefore, I ask you to very carefully and in detail correct the article and respond to my comments.
Comments on the Quality of English Language
Stylistic, technical and grammatical corrections are needed.
Reviewer 3 Report
Comments and Suggestions for Authors
Dear author,
the manuscript is generally well-written, however some correction and improvements are needed.
Following you can find the comments point by point
General comments:
- Add the space before every citation
- Check along the text that Quaidam Basin is always spelled the same (upper and lower case letters)
Simple summary:
- Line 14: Please specify the country of the six regions of Quaidam Basin
Abstract:
- Lines 33-34: specify the acronym of the groups
Introduction:
The introduction needs to be improved. References to the state of the art regarding the study of microbiota in gazelles and/or wild animals should be added. Soil characteristics that may influence the microbiota should also be more specified
Materials and Methods:
- Line 92: Please put "th" as superscript
- Lines 92-93: Please provide the entire name out of the brackets while the acronym will be putted inside the brackets
- Lines 119 - 122: Pleae provide the reference for the primer
- Line 156: delete the dot between "reduced" and "except"
Results:
- In general when the mean is reported you have to present also the standard deviation
- Figure 3A: Please revise the Y axes, since is no possible to identify the column related to "Protobacteria" and "Cyanobacteria"
- Figure 5: Please revise the Y axes, since is no possible to identify all the column
- Figure 6: Significances represented in this way are difficult to understand. Please revise them
- Figure 7: Significances represented in this way are difficult to understand. Please revise them
Discussion:
- Line 298: Please change "results" with "discussion"
- Line 362: Please specify at this point the acronym F/B
Round 2
Reviewer 1 Report
Comments and Suggestions for Authors
Dear Authors,
The correction of the manuscript corresponds to my ideas. The introduction of changes has improved the manuscript. Certainly, the changed title of the manuscript brings clarity. The addition of illustrative material and additions to the text made the manuscript raise the qualitative level of work. The review of the research results was chosen appropriately, as were the statistical methods used for its analysis. The article takes into account the comments on the methodology. The analysis and conclusion for each chapter are sufficient and do not raise objections. References to sources of literature have been adjusted. The results of previous studies by other authors have been taken into account. I recommend it for the journal Animals.
Reviewer 2 Report
Comments and Suggestions for Authors
Dear authors,
Thank you for your responses to my comments. I have two more comments regarding the additional OTU table. I would like to ask you to add the taxonomic assignment of each OTU and calculate the relative abundance in percent.
My second question is that if you included archaea in the analysis, why is there no mention of them in the text? As I mentioned in the first round of review, archaea play an important role in the gastrointestinal microbiome of ruminants. Please, reassess this point.
Reviewer 3 Report
Comments and Suggestions for Authors
The authors fully responded to the comments made, consequently improving the submitted manuscript. Therefore, I believe it can be considered for publication
